# Chemical Composition and Biological Activities of Essential Oils from *Origanum vulgare* Genotypes Belonging to the Carvacrol and Thymol Chemotypes

**DOI:** 10.3390/plants12061344

**Published:** 2023-03-16

**Authors:** Paola Zinno, Barbara Guantario, Gabriele Lombardi, Giulia Ranaldi, Alberto Finamore, Sofia Allegra, Michele Massimo Mammano, Giancarlo Fascella, Antonio Raffo, Marianna Roselli

**Affiliations:** 1CREA-Research Centre for Food and Nutrition, Via Ardeatina, 546, 00178 Rome, Italy; 2Institute for the Animal Production System in the Mediterranean Environment, National Research Council, P.le E. Fermi 1, 80055 Portici, Italy; 3Department of Environmental Biology, Sapienza University, P.le Aldo Moro 5, 00185 Rome, Italy; 4CREA-Research Centre for Plant Protection and Certification, S.S. 113-Km 245.500, 90011 Bagheria, Italy

**Keywords:** oregano, *Origanum heracleoticum*, *Origanum vulgare ssp. viridulum* × *Origanum vulgare ssp. hirtum*, essential oil, enantiomers, antimicrobial activity, Caco-2 cells, intestinal permeability, reduction in pathogen adhesion, geographic origin

## Abstract

The remarkable biological activities of oregano essential oils (EOs) have recently prompted a host of studies aimed at exploring their potential innovative applications in the food and pharmaceutical industries. The chemical composition and biological activities of EOs from two *Origanum vulgare* genotypes, widely cultivated in Sicily and not previously studied for their biological properties, were characterized. Plants of the two genotypes, belonging to the carvacrol (CAR) and thymol (THY) chemotypes and grown in different cultivation environments, were considered for this study. The chemical profiles, including the determination of enantiomeric distribution, of the EOs, obtained by hydrodistillation from dried leaves and flowers, were investigated by GC–MS. Biological activity was evaluated as antimicrobial properties against different pathogen indicator strains, while intestinal barrier integrity, reduction in pathogen adhesion and anti-inflammatory actions were assayed in the intestinal Caco-2 cell line. The chemical profile of the CAR genotype was less complex and characterized by higher levels of the most active compound, i.e., carvacrol, when compared to the THY genotype. The enantiomeric distribution of chiral constituents did not vary across genotypes, while being markedly different from that observed in *Origanum vulgare* genotypes from other geographical origins. In general, all EOs showed high antimicrobial activity, both in vitro and in a food matrix challenge test. Representative EOs from the two genotypes resulted not altering epithelial monolayer sealing only for concentrations lower than 0.02%, were able to reduce the adhesion of selected pathogens, but did not exert relevant anti-inflammatory effects. These results suggest their potential use as control agents against a wide spectrum of foodborne pathogens.

## 1. Introduction

Oregano is one of the most commercially valued aromatic plants worldwide, with more than 60 species used under this name having similar flavor profiles characterized mainly by cymyl compounds, such as carvacrol and thymol [1]. Beyond the widespread use of oregano as a culinary herb, its essential oil (EO) is currently used as an ingredient in flavoring formulations, dietary supplements and cosmetic and aromatherapeutic products [2]. A great deal of research has been recently dedicated to exploring other potential applications of oregano EOs in the food industry as food preservatives, based on their remarkable antimicrobial and antioxidant properties [3]. When considering such potential applications, it is important to take into account that foodborne diseases represent a serious and widespread threat to public health worldwide and require the adoption of strategies that ensure food safety and quality. An analysis of the European Rapid Alert System for Food and Feed (RASFF) Annual Report in 2020 [4] showed that a high number of notifications concerned microbiological contamination from pathogenic microorganisms in food of mostly animal origin, with a 37% increase in notifications in 2020 compared to 2019, when the largest number concerned *Salmonella* spp., *Listeria monocytogenes* and *Escherichia coli*. Concerning other potential applications in food production, a more recent field of research has focused on the use of oregano EOs as environmentally friendly alternatives to synthetic antibiotics in the aquaculture industry [5] or to synthetic pesticides as innovative products for crop protection [6]. Moreover, the potential of oregano EO as a protective agent in human chronic degenerative and infectious diseases, suggested by its several biological activities such as antimicrobial, antifungal, antiparasitic, anti-inflammatory, anticancer, antiproliferative, cytotoxic, has prompted a host of studies related to its potential pharmaceutical applications [7,8,9]. In particular, microbial resistance to antibiotics and chemotherapeutics is rapidly growing [10,11], requiring the search for alternative sources of antimicrobial compounds, both for prevention in human and veterinary medicine and for food preservation. Over the centuries, plants have been used for a wide variety of purposes, ranging from the treatment of infectious diseases to the extension of food shelf life, and been considered important sources of compounds with great chemical diversity, reflecting different biological properties [12]. In this context, oregano is considered among the plants whose extracts have the greatest antimicrobial effects [13]. Thus, a lot of efforts are currently underway to develop micro- and nano-encapsulation systems that enable the use of oregano EO in biotechnological and biomedical applications, by increasing its stability in aqueous media, thus improving its bioavailability, reducing its toxic effects, providing a controlled release and masking its strong aroma [14,15].

EOs obtained from *Origanum vulgare*, which is the main representative species of the genus *Origanum* L., are formed by a complex mixture of terpenes. Based on their proportions of cymyl, acyclic linalool/linalyl acetate and sabinyl compounds, three major chemotypes have been identified [16]. Within this complex chemical mixture, the two structurally similar cymyl compounds, carvacrol and thymol, are recognized as the main ones responsible for most of the biological activities of oregano EOs, and in particular, most of their antimicrobial activities [17]. It is generally assumed that their antibacterial action is exerted by inducing structural and functional damages to the cytoplasmic membrane of the target organism. However, a lot of evidence indicates that the biological activities of an EO may depend not only on the ratio in which the main active compounds are present but also on interactions between these and minor constituents in the oil [17]. From this perspective, it may be important to explore the chemical variability of oregano EO composition and to understand the influence of this variability on its biological activities.

One of the main sources of chemical variability in oregano EO is related to the genetic background of the plant. From this point of view, the species *Origanum vulgare*, native to and widespread in the Mediterranean region [18], encompasses many subspecies and hybrids, among which is the subspecies *Origanum heracleoticum* L. (Boissier) Hayek sensu Ietswaart [19], which is one of the most common growing in Sicily [20]. Information is available on the phytochemistry of this subspecies [20,21], but the biological activities of its EOs have not yet been investigated. On the contrary, no studies to the best of our knowledge have been conducted both on the chemical and biological features of the hybrid *Origanum vulgare ssp. viridulum* × *Origanum vulgare ssp. hirtum*, introduced into cultivation in Sicily in the last decade.

A lot of investigations have explored the intraspecific variability of the chemical composition of EOs from *Origanum vulgare* species, while a remarkable variation has also been observed within each subspecies, bearing to noticeable differences in the chemical profile and, possibly, in the associated biological properties [20,22]. Beyond genetic factors, this variability may also be due to environmental factors, such as, among others, geographical position [23] and altitude [24]. Regarding the chemical characterization of oregano EOs, very few data are available on the enantiomeric distribution of their main chiral compounds [25], even though this information may be useful not only for chemotaxonomical characterization, but also for authenticity issues. Changes in enantiomeric distribution could also influence the biological activities of oregano EO [26].

In the present study, a detailed characterization of the chemical composition and biological activities of EOs obtained from plants of the subspecies *Origanum heracleoticum* L. and the hybrid *Origanum vulgare ssp. viridulum* × *Origanum vulgare ssp. hirtum* is provided. The chemical characterization, besides their basic chemical profile, also included the determination of the enantiomeric distribution of chiral compounds present in their EOs and in their dried leaves and flowers. The study of their biological activities focused on antimicrobial, intestinal permeability and anti-inflammatory properties. To evaluate antimicrobial properties, different pathogen indicator strains were used, while the intestinal barrier integrity, reduction in pathogen adhesion and anti-inflammatory activity were evaluated in a widely used in vitro model of intestinal cells, the human Caco-2 cell line, differentiated on permeable filters. These conditions allowed for the reproduction of the environment of the intestinal mucosa, in which the epithelium separates the lumen from the inner body. A preliminary characterization carried out in our laboratory highlighted that the hybrid *Origanum vulgare ssp. viridulum* × *Origanum vulgare ssp. hirtum* was distinguished by a high content of carvacrol as the main EO constituent, thus allowing for a description of its EO as the carvacrol chemotype, whereas EOs from the subspecies *Origanum heracleoticum* L. are known to belong to the thymol chemotype [20]. Thus, the aim of the present study was to provide detailed information on the chemical and biological properties of the EOs from these two genotypes to promote alternative and innovative applications in food and pharmaceutical products suited to their specific properties. In addition, the information collected may contribute to a deeper understanding of the influence of varying chemical profiles, such as those with carvacrol or thymol as predominant constituents, on the biological activities of oregano EOs.

## 2. Results

### 2.1. Composition of EOs

GC–MS analyses enabled the detection of 32 volatile compounds at a level higher than 0.05% of total EO content (Table 1) in three EOs obtained from the same genotype of the hybrid *Origanum vulgare ssp. viridulum* × *Origanum vulgare ssp. hirtum* (CAR1, CAR2, CAR3) and in five EOs distilled from plants from the same genotype of the subspecies *Origanum heracleoticum* L. (THY1, THY2, THY3, THY4, THY5). Within each group of EOs, the plants obtained from the same genotype were grown on different farms located in a restricted area of southern Sicily. Thirty-one of the detected compounds were identified as monoterpene hydrocarbons (thirteen), oxygenated monoterpenes (eight), sesquiterpene hydrocarbons (eight) and two other compounds. The main feature of both groups of EO samples was the occurrence of carvacrol or thymol as the predominant constituent: in the EOs from the hybrid *Origanum vulgare ssp. viridulum* × *Origanum vulgare ssp. hirtum*, named the CAR group, carvacrol was the prominent compound, varying in amount from 81 to 85%, whereas in the EOs from the subspecies *Origanum heracleoticum* L., named the THY group, the main constituent was thymol, ranging in amount from 47 to 65%. Globally, the group of monoterpene hydrocarbons accounted from a higher proportion of EO content in the THY group (24–38%) than in the CAR group (12–14%): in both groups, γ-terpinene and p-cymene were the main constituents of the monoterpene hydrocarbon chemical class, being present at a higher level in the THY group (13–22% and 4–5% in THY, and 5–7% and 3% in CAR, respectively). Similarly, the sesquiterpene hydrocarbon content was higher in the THY group (3.9–6.9%) than in the CAR group (2.2–2.6%), with β-caryophyllene occurring as the main constituent (0.9–1.8% in THY and 1.9–2.4% in CAR). Conversely, the global level of oxygenated monoterpenes showed a higher content in the CAR group (84–85%) than in the THY group (56–71%), this difference being dictated by the level of carvacrol/thymol. Interestingly, when 0.05% was established as the cut-off level for quantification, EOs of the THY group were characterized by a more complex chemical profile, formed by 30 compounds, whereas only 19 compounds were represented in the CAR group, in which a larger proportion of total content was formed by a single compound. Moreover, the chemical profiles of the EOs of the CAR group were quite similar to each other, whereas a higher level of variability was observed in the profile of the EOs of the THY group, suggesting a higher influence of the farm location on the chemical profile for this group (Figure 1). Diversification of chemical profiles in the THY group was mainly due to variability in the level of monoterpene hydrocarbons, such as α- and β-pinene and α-thujene (Figure 1).

### 2.2. Enantiomeric Distribution of Chiral Compounds as Determined after Isolation from Oregano Dried Leaves and Flowers and in Distilled Oregano EOs

The enantiomeric distribution analysis of chiral volatile compounds of oregano leaves and flowers was performed both on the volatile isolates directly obtained by headspace solid-phase microextraction (HS-SPME) from the dried plant material (Table 2a) and on the corresponding EOs obtained from the plant material by distillation (Table 2b). GC analysis performed by an enantioselective GC column allowed for the detection of seven pairs of enantiomers in four monoterpene hydrocarbons, α-thujene, α-pinene, β-pinene, α-phellandrene and three oxygenated monoterpenes, linalool, terpinen-4-ol and α-terpineol. This last compound was not quantified in the GC analysis with an achiral column (Table 1) because its level was lower than 0.05%, but it was included in the enantioselective analyses because it was possible to accurately determine its two enantiomers. The analysis of the isolates obtained from the dried leaves and flowers by HS-SPME allowed for the determination of a higher number of enantiomeric pairs than in the analysis of the EOs, because the former isolates were more concentrated. In the case of α-thujene, it was not possible to identify the two enantiomers, due to unavailability of commercial standard pure compounds and lack of appropriate information from the literature of chromatographic data. The results obtained for both the dried aerial parts and EOs showed that the enantiomeric distribution was the same in plants belonging to the subspecies *Origanum heracleoticum* L. and the hybrid *Origanum vulgare ssp. viridulum* × *Origanum vulgare ssp. hirtum*, and it was not affected by the different environments of cultivation. Only in the case of the distribution of linalool enantiomers, as determined in the dried plant materials in one sample (CAR3), was there a slight difference in the distribution of enantiomers observed, with a 17% level of the S enantiomer instead of the 8–11% level observed in the other samples. Moreover, results obtained from the dried plant material showed less variability when compared to those obtained from the EOs. This could partly be due to the more concentrated isolate analyzed in the former case. Enantiomeric distribution as determined in the dried plant material was generally similar to that observed in the EOs, with some minor differences. For instance, the distribution of α-pinene enantiomers was, on average, 5:95 and 10:90 in the EOs and plant material, respectively; the distribution of β-pinene was 81:19 and 73:27; the distribution of terpinene-4-ol was 61:39 and 51:49; and the distribution of α-terpineol was 0:100 and 9:91. In all cases, a slightly higher enantiomeric excess was observed in the EOs than in the isolates from dried plant material.

### 2.3. Antimicrobial Activity of EOs by Agar Spot Test against Pathogen Indicator Strains

EOs extracted from the hybrid *Origanum vulgare ssp. viridulum* × *Origanum vulgare ssp. hirtum* showed a relevant antimicrobial activity assayed by an agar spot test, with halos of various sizes (Table 3). In particular, all the pathogenic microorganisms considered were susceptible to the inhibitory activity of CAR1, CAR2 and CAR3, with inhibition halo radii exceeding 7.5 mm, with the exception of *L. monocytogenes* SA and *Pseudomonas fluorescens* B13, against which these oils had low activity.

Concerning the EOs extracted from *Origanum heracleoticum,* THY3 and THY5 were tested, as they resulted as more divergent within the THY cluster (Figure 1). These two EOs showed a medium activity against *P. putida* KT2240 and *P. fluorescens* B13 (radii between 5 and 7.5 mm) and high inhibitory activity against all other pathogenic and alterative microorganisms (Table 3).

On the basis of the results of the chemical characterization (Table 1) and antimicrobial activity by spot test (Table 3), CAR1 and THY5 were selected for further experiments, as they had higher carvacrol and thymol content among CAR1, CAR2 and CAR3 and among THY3 and THY5, respectively. Moreover, CAR1 and THY5 showed high antimicrobial activity against *L. monocytogenes* OH and *S.* Typhimurium LT2 and were thus chosen as representative of the most frequently reported pathogen related to foodborne diseases.

### 2.4. In Vitro Antimicrobial Activity of CAR1 and THY5 against L. monocytogenes OH and S. Typhimurium LT2 by Direct Contact Test

The two EOs CAR1 and THY5 showed similar antimicrobial capacities by a direct contact test against *L. monocytogenes* OH (Figure 2A). In particular, at a lower EO concentration (0.12%), an approximately 3-log reduction in microbial concentration occurred as compared to samples exposed to 0% EO (C) and to 0% EO plus 0.125% Tween 80 (CTw). Proportionally to increased EO concentration (0.25 and 0.5%), a reduction of approximately 4 and 4.5 logs, respectively, was observed in the microbial titer, as compared to C. In addition, the treatment with CAR1 and THY5 at 0.25% induced a reduction in microbial concentration similar to that obtained with ampicillin, while for both oils, treatment with 0.5% induced a greater effect than ampicillin in reducing microbial load (Figure 2A). Similarly, the oils at the same extent showed antimicrobial activity by a direct contact test against *S.* Typhimurium LT2 (Figure 2B), which resulted in a reduction in the microbial titer of about 0.5 log CFU/mL when applied at a concentration of 0.12%. A dose–response effect was observed, as the 0.5% concentration induced a reduction of about 4 log CFU/mL. For *S.* Typhimurium LT2, the antimicrobial effects of the oils at a 0.5% concentration was similar to that of ampicillin (Figure 2B).

### 2.5. Antimicrobial Effects of CAR1 and THY5 during a Challenge Test of Beef Minced Meat

The antimicrobial effects of CAR1 and THY5 EOs against *L. monocytogenes* OH and *S.* Typhimurium LT2 was also evaluated in a complex food matrix of beef minced meat during storage at 4 °C for 7 days. The 0.5% concentration was used in order to reproduce the conditions of maximum antimicrobial concentration as in the direct contact test. The presence of CAR1 EOs caused a reduction of the bacterial population of both *L*. *monocytogenes* OH and *S.* Typhimurium LT2 by about 1 log CFU/g one day after contamination, remaining essentially unchanged until the third day (Figure 3A,B). In both sets of experiments, there was an increase in microbial load after seven days of storage both in the presence and absence of the EOs, suggesting an onset of recontamination of the minced meat. In parallel, the absence of *L. monocytogenes* OH and *S.* Typhimurium LT2 in the food matrix in the artificially uncontaminated sample was assessed.

Regarding the antimicrobial activity of 0.5% THY5, a different effect was observed against the two pathogens tested. The microbial load of *L. monocytogenes* OH remained essentially unchanged after 1 and 2 days in the contaminated samples in the presence of EO, whereas a reduction of approximately 1 and 2 logs in the microbial load was observed at 3 and 7 days, respectively (Figure 3C). The minced meat in the absence of THY5 had a similar microbial titer of about 1 × 10^5^ CFU/g during all storage timepoints.

The results of the challenge test set up with *S.* Typhimurium LT2 indicated that the antimicrobial effect of THY5 EO was already appreciable at the first day with a reduction of approximately 1.5 log CFU/g and a further decrease of 1 log CFU/g was observed at the following timepoints (2, 3 days). On day 7, an increase of about 0.6 log CFU/g suggested the start of recontamination of the minced meat (Figure 3D). Surprisingly, from the second day of storage at 4 °C, a reduction in the microbial titer was observed in the absence of EO.

### 2.6. Effects of CAR1 and THY5 on Caco-2 Cell Permeability

In order to evaluate if EOs exposure could perturb intestinal epithelial permeability, transepithelial electrical resistance (TEER) and phenol red apparent permeability (Papp) were measured in differentiated Caco-2 cells after treatment with several concentrations (ranging from 0.01 to 0.1%) of the two representative EOs CAR1 or THY5, dissolved in a complete culture medium for up to 24 h. The results show that only the 0.01 and 0.02% CAR1 treatments did not affect cell permeability up to 24 h, while the higher concentrations (0.025–0.03–0.05–0.1%) already induced a TEER drop after 1 h (less than 20% of the control), which was maintained until the end of the experiment (Figure 4A). This TEER decrease was associated with a biologically relevant phenol red Papp increase, as the corresponding values were in the order of magnitude of 6 × 10^−6^ cm s^−1^, indicating that the tight junctions were open (Figure 4B). Indeed, permeability coefficients higher than 5 × 10^−7^ cm s^−1^ are considered to be indicative of compromised cell monolayers, while values above 1 × 10^−6^ cm s^−1^ indicate destroyed cell monolayer integrity [27]. Concerning THY5, a drop in TEER values was already observed after 1 h treatment with 0.03, 0.05 and 0.1% concentrations (Figure 4C). Differently from CAR1, the 0.025% concentration of THY5 induced a TEER decrease less rapidly, but regardless, was indicative of important and irreversible damage to the monolayer, since from 4 h treatment, the TEER values were lower than 50% of the control and they continued to decrease. At the lower concentrations tested (0.02 and 0.01%), the TEER values remained above 100% of the control up to 24 h, indicating the integrity maintenance of the monolayer (Figure 4C). The TEER data were confirmed by the results of the paracellular passage of phenol red, shown in Figure 4D. The Papp values were indeed higher than the threshold value (1 × 10^−6^ cm s^−1^) for THY5 concentrations between 0.1 and 0.025%, while those related to the treatment with 0.02 and 0.01% were in the order of 1 × 10^−7^ cm s^−1^, indicating that the tight junctions between cells were functionally sealed. From these data, it appears that among the concentrations tested, the highest concentration not damaging the Caco-2 cell monolayer was 0.02% for both EOs. This concentration was therefore used for the experiments of reduction in pathogen adhesion and for the evaluation of anti-inflammatory activity. The presence of 2% ethanol, corresponding to the concentration contained in the higher EO dilution tested, did not affect cell permeability for up to 24 h.

### 2.7. Reduction in Pathogen Adhesion to Caco-2 Cells by CAR1 and THY5

To evaluate the potential reduction ability exerted by 0.02% CAR1 and THY5 against pathogen adhesion on Caco-2 cells, the indicator strains *L. monocytogenes* OH and *S.* Typhimurium LT2 were selected. Both strains had a similar ability to adhere to Caco-2 cells, quantified in about 10% of the initial inoculum, which consisted of approximately 1 × 10^8^ CFU/mL for each strain (Figure 5A,B). The presence of CAR1 significantly reduced the adhesion of *L. monocytogenes* OH as compared to the oil-free controls, with a decrease of approximately 1 log CFU/mL, while THY5 was not effective (approximately 0.5 log CFU/mL, Figure 5A).

Both CAR1 and THY5 were able to significantly reduce the number of adhered viable bacterial cells of *S.* Typhimurium LT2 to Caco-2 cells (Figure 5B). Thus, the two pathogens tested showed a different susceptibility to CAR1 and THY5.

Such adhesion reduction was not ascribable to the presence of 0.4% ethanol, as preliminary experiments did not show any effect of this ethanol concentration (contained in 0.02% EOs) on the number of viable bacterial cells adhering to the intestinal cell monolayer.

### 2.8. Effects of CAR1 and THY5 on the NF-kB Pathway in Caco-2 Cells

To evaluate if CAR1 and THY5 could have a protective effect on a pro-inflammatory stimulus (TNF-α), the gene expression of some key genes of the NF-kB pathway was analyzed in Caco-2 cells. Epithelial integrity was not affected by experimental treatments with TNF-α or EOs, as indicated by the TEER values that remained comparable to the untreated control cells throughout all the experimental intervals.

As compared to the control, treatment with CAR1 induced a mild gene expression downregulation of the IkBα and IL-6 genes of the NF-kB pathway; however, this downregulation did not reach statistical significance, probably due to great data variability (Figure 6A). THY5 treatment did not affect gene expression (Figure 6B).

As expected, TNF-α treatment was able to induce a significant increase in the expression (up to 2 fold change) of all analyzed genes of the NF-kB pathway, except for IL-1α (Figure 6A,B). CAR1 pre-treatment before TNF-α addition induced a slight decrease in IkBα, IL-6 and IL-8 gene expression as compared to treatment with TNF-α alone. However, this remained as a trend (*p* > 0.05), as statistical significance could not be reached because of great data variability (Figure 6A). When the cells were pre-treated with THY5 before TNF-α addition, the gene expression levels of all analyzed genes were similar to TNF-α, even if an increasing trend could be observed for inflammatory IL-1α, IL-6 and IL-8 cytokines (Figure 6B).

## 3. Discussion

In the present study, eight different EOs distilled from plants obtained of two genotypes belonging to the subspecies *Origanum heracleoticum* L. and to the hybrid *Origanum vulgare ssp. viridulum* × *Origanum vulgare ssp. hirtum* were characterized according to their chemical composition and several biological activities. First, chemical analyses allowed for highlighting the differences in the profiles of the two genotypes and evaluating the variability within each genotype depending on the cultivation location. EOs from the two genotypes showed differences not only in their predominant compound, carvacrol or thymol, but also in their overall profile complexity, which was more pronounced in the thymol chemotype. The profile of the EOs from the *Origanum heracleoticum* L. subspecies, described as belonging to the thymol chemotype, was in accordance with the data reported in the literature on the same subspecies of wild and cultivated oregano plants grown in the same geographic area of Sicily [20,21,28]. No previous information was available in the literature, to the best of our knowledge, on the profile of the hybrid *Origanum vulgare ssp. viridulum* × *Origanum vulgare ssp. hirtum*, described as belonging to the carvacrol chemotype. In both groups, the occurrence of relatively high levels of γ-terpinene and p-cymene was consistent with the biosynthetic pathway proposed for the main compounds carvacrol/thymol, by which γ-terpinene is first converted to p-cymene and, in turn, further converted to carvacrol or thymol by two distinct enzymes [29]. Interestingly, while most monoterpene and sesquiterpene hydrocarbons appeared to be positively correlated among themselves and with thymol, β-caryophyllene was positively correlated with carvacrol and negatively correlated with the other sesquiterpenes. From the point of view of relationships with biological activities, it is interesting to note that the level of the most active EO compound was significantly higher in the hybrid *Origanum vulgare ssp. viridulum* × *Origanum vulgare ssp. hirtum*. than in the subspecies *Origanum heracleoticum* L., which might suggest a higher potential in the former. On the contrary, the profile of EOs from the subspecies *Origanum heracleoticum* L. was more complex than the other in terms of the number of minor constituents detected. Moreover, chemical characterization provided detailed information on the enantiomeric distribution of chiral compounds in the aerial parts of the considered oregano genotypes and in the corresponding EOs. This piece of information may be important not only because optical isomers can have different odors and, in general, different biological properties, but also because it provides useful information for uncovering the biosynthetic and geographical origins of the considered EOs as well as their authenticity and also for investigating technological treatments to which EOs could be subjected [30]. In the oregano EOs analyzed in this study, chiral compounds were all present at a relatively low level (<1%), whereas all major compounds were achiral. This would suggest a minor relevance of the enantiomeric characterization for these oregano genotypes. However, very little information is available in the literature on the enantiomeric distribution of chiral compounds in oregano EOs, and the only paper reporting experimental data showed a quite different picture as compared to our results, both in terms of chemical profiles and enantiomeric distributions [25]. In that study, EOs from native populations of *Origanum vulgare* collected in two different geographical locations in northwestern Himalaya and belonging to the linalool and cymyl chemotypes were investigated [25]. In that study, pairs of enantiomers were determined for four of the seven chiral compounds investigated in the present study, i.e., α-pinene, linalool, terpinene-4-ol and α-terpineol. In all cases, the enantiomeric distributions observed were markedly different between the two populations from Himalaya. Thus, the determination of enantiomeric distribution was confirmed to represent a valuable analytical tool for a detailed chemical characterization of oregano EOs, which may be useful for the recognition of geographical origin. Moreover, in that study, a marked effect of the distillation process on the observed enantiomeric distribution in the obtained EOs was postulated to explain some inversions of the ratios for some compounds between the two EOs investigated [25]. Data from the present study seemed to rule out the possibility of a marked effect of the distillation step on the enantiomeric distributions, while the minor differences observed between the EOs and dried aerial parts could be plausibly attributed mainly to differences in the analytical process, and, in particular, to the different concentration levels of individual VOCs in the isolates analyzed in the two cases. Moreover, an expected effect of the distillation process could be a higher extent of racemization, whereas on the contrary in this study, a slightly lower extent of racemization was observed in the EOs instead of the dried plant material.

On the other hand, the analysis performed on several biological activities of the EOs from the two genotypes provided a characterization useful for highlighting their possible distinguishing properties among different oregano genetic resources and for evaluating their potential for food and pharmaceutical applications.

All the EOs showed a marked dose-dependent antimicrobial activity against some pathogenic and spoilage bacteria, belonging to both Gram-negative and Gram-positive groups. In the direct contact in vitro test performed with CAR1 and THY5, a greater effect was observed against *L. monocytogenes* OH, a Gram-positive species, as compared to the Gram-negative *S.* Typhimurium LT2. These data are coherent with the antibacterial mechanism elucidated for thymol and carvacrol: these oxygenated monoterpenes, bearing a phenolic functional group, are able to interfere with the phospholipid bilayers, damaging the bacterial cell membrane and causing a decrease in ATP levels until the cell dies [3,31,32], while the hydrophilic nature of the outer membrane of Gram-negative bacteria exerts a shielding action towards EOs [33,34].

The antimicrobial activity of CAR1 and THY5 was also observed during a challenge test, in which a food matrix of minced beef was used to study the behaviour of a pathogenic or spoilage microorganism under specific storage conditions [35]. Although both CAR1 and THY5 showed antimicrobial activity, THY5′s effect appeared to be slower and more persistent than CAR1′s in the food contaminated by *L. monocytogenes* OH. The different dynamics of the two pathogens observed in this assay in the presence and absence of oil treatment could be partly ascribable to an inherent variability and complexity in the food matrix. The increase in microbial load observed after seven days of storage both in the presence and absence of the EOs could be due to multiple concomitant causes: e.g., recontamination of the ground meat or a decrease in the EO activity due to too long of a storage time.

Another Interesting aspect to consider in terms of antimicrobial activity is the effect of the minor components present in EOs [3]. Sivropoulou et al. [36] investigated the antimicrobial effect of EOs extracted from *Origanum vulgare ssp. hirtum*, *Origanum dictamnus* and a commercial oregano EO, concluding that these oils, although different in chemical composition, exhibited similar antibacterial activities. The authors attributed this behaviour to the action of minor compounds that could have their own activity or act in synergy or antagonism with the main ones. Several studies have reported synergistic activity among EO components: for example, the presence of carvacrol and *p*-cymene causes a major destabilization of the bacterial membrane and a reduction in membrane potential [37,38]. The results of the present study confirmed that the diversity in the chemical profile between the two genotypes did not produce marked differences in antibacterial activity.

The potential applications of EOs as food preservatives as well as food ingredients cannot be separated from the evaluation of their impact on intestinal health following their intake. Thus, in this study, we investigated possible adverse and beneficial actions of EOs on the intestinal mucosa since it represents the first site of impact between ingested food and the organism. To such purposes, human intestinal epithelial Caco-2 cells were used from a well characterized enterocyte-like cell line capable of expressing the morphological and functional differentiation features typical of mature enterocytes, including cell polarity and a functional brush border [39]. The Caco-2 cell line has been extensively used as a reliable in vitro system to study cell toxicity, inflammation and intestinal injury induced by pathogens [40].

The effect of oregano EOs on intestinal cells is quite unexplored, except for a couple of papers evaluating antioxidant activities [41,42]; yet, the few available studies are focused on isolated components, such as thymol and/or carvacrol, with a particular emphasis on cell toxicity [43]. Another study conducted on undifferentiated Caco-2 cells treated up to 48 h with relatively high concentrations (2.5 mM and below) of carvacrol and thymol demonstrated heavy perturbation of the cellular ultrastructure and apoptosis [44]. However, it should be emphasized that the activity of pure compounds (singly or in mixtures) can be very different from that of complete EOs, which also contain a variety of secondary compounds.

The incubation of Caco-2 cells in the presence of both CAR1 and THY5 caused a significative rapid and strong reduction in TEER values accompanied by a high passage of the paracellular marker phenol red at the higher concentrations tested (0.1%), approximately corresponding to 5.2 mM of carvacrol and 3.6 mM of thymol for CAR1 and THY5, respectively. Although this trend was relatively less sharp for the THY treatment at a 0.025% concentration, this sudden and strong toxic response likely indicated cell death occurrence, suggesting unspecific cytotoxicity effects, probably due to EO’s ability to permeabilize mitochondrial membranes leading to cell death [45]. This was also confirmed by Fabian and co-workers [46], who observed a high necrosis rate in Caco-2 cells treated for 1 h with oregano EO at concentrations comparable to ours. However, at concentrations lower than 0.02%, both CAR1 and THY5 did not affect Caco-2 monolayer integrity for up to 24 h of incubation, indicating a complete absence of adverse effects under these experimental conditions and likely suggesting a level of safe concentrations at which EOs do not alter intestinal integrity. To the best of our knowledge, this is the first study evaluating the intestinal permeability effects of oregano EOs in human intestinal cells. These in vitro tests provide early and specific intestinal toxicity assays for identifying dietary compounds that may affect intestinal barrier function and represent a valuable tool for further characterizing oregano EO safety in humans, as already shown with oral toxicity tests performed on rats [47].

Data reported in the literature relating to oregano EO anti-inflammatory activity suggest their ability to reduce inflammation in several in vitro and in vivo models [48,49,50]; however, in our intestinal Caco-2 cell system, pre-incubation with EOs did not induce evident protection from TNF-α inflammatory challenge. Nevertheless, CAR1 preincubation followed by TNF-α exposure determined a weak downregulation of genes involved in the NF-kB pathway. This trend might confirm potential CAR1 anti-inflammatory activity in this experimental system, which could not be further assayed at a higher CAR1 concentration due to its toxicity. Finally, neither CAR1 nor THY5 displayed inflammatory action on their own, as incubation with these EOs did not induce any significant increase in the expression of the inflammatory genes tested, as compared to the control cells.

As reported by Di Vito et al. [31], the initial stages of pathogen adhesion to intestinal cells are crucial for the establishment of colonization, with the risk of being a reservoir of acute events; thus, the search for new inhibitors of bacterial adhesion to host tissues turns out to be very important. The results of the pathogen adhesion assay, described in the present work, showed that the addition of CAR1 and THY5 oils, despite being at low concentrations, was indeed able to reduce bacterial adhesion to the intestinal Caco-2 cell line, although THY5 was effective only against *S.* Typhimurium LT2. Our data are comparable to those of other research carried out on thyme EOs, in which both thymol (about 12%) and carvacrol (about 68%) were present; in particular, it was observed that both *Thymus capitatus* EO as well as thymol and carvacrol alone were able to reduce the adhesion of *E. coli* (ATCC35210) and *L. monocytogenes* (NCTC 7973) to the intestinal HT-29 cell line [51]. Previous work has demonstrated that carvacrol was able to reduce *Campylobacter jejuni* infection in epithelial cells [52], while thymol decreased the adhesion of various *E. coli* and *S. aureus* strains on vaginal epithelial cells [53]. Finally, both compounds were able to reduce the adhesion of three *L. monocytogenes* strains to Caco-2 cells [54]. Furthermore, in a recent study carried out on *Origanum vulgare* EO, a reduction in the adhesion of several *Salmonella* serotypes to Caco-2 cells was observed by a commercial mixture of EO containing both thymol and carvacrol; this effect was suggested to occur through the inhibition or disaggregation of bacterial biofilm [31]. The mechanisms underlying this effect can be multiple and ascribable to the antimicrobial activity of EOs against Gram-positive and Gram-negative pathogens, to their ability to alter the chemical–physical properties of the bacterial surface or microbial motility, as well as to the reduction in gene expression of some virulence factors [52,53,54]. Further experiments will be needed to fully understand the mechanisms underlying the protective effects exerted by oregano EOs.

## 4. Materials and Methods

### 4.1. Plant Materials

The plant material used for this study was obtained from eight farms located in southwestern Sicily, in different areas of the province of Agrigento, at the following altitudes: farm 1 at 250 m a.s.l., 2 at 300 m, 3 at 200 m, 4 at 250 m, 5 at 300 m, 6 at 150 m, 7 at 250 m and 8 at 250 m. Plants from the hybrid *Origanum vulgare ssp. viridulum* × *Origanum vulgare ssp. hirtum* were provided by farms 1–3 (indicated in the text as CAR1, CAR2, CAR3), whereas plants from the subspecies *Origanum heracleoticum* L. were provided by farms 4–8 (THY1, THY2, THY3, THY4, THY5). It is not clear whether the subspecies *Origanum heracleoticum* L. should have been identified as *Origanum vulgare ssp. viridulum* or as *Origanum vulgare ssp. hirtum*, with contrasting identifications having been reported in the literature [20,21]. For both groups of oregano types, all plants were obtained by the propagation of the same genotype and then grown in different cultivation environments, i.e., the eight farms. On all the farms, oregano was grown according to the cultivation practice and crop management traditionally applied in these areas of Sicily [20]. Sampled plants were harvested in mid-June by cutting up to 5 cm above the soil level and then air-dried in the shade at an air temperature ranging between 25 and 30 °C, for about 10–15 days. The dried material was shipped to the CREA-Research Centre for Food and Nutrition laboratory, where all the analytical determinations were carried out.

### 4.2. Isolation of the EOs

The samples of air dried leaves and flowers (20 g) were subjected to hydrodistillation by a Clevenger-type apparatus for 3 h and then, the collected EO was dried over anhydrous sodium sulphate and stored under N_2_ in a sealed vial at −24 °C until analysis.

### 4.3. GC–MS Analyses of EO Compositions

The EOs were diluted in methanol (1:20 *v*/*v*) before the GC analysis. GC analyses were performed on an Agilent 6890 5973 N, GC–MS system (Agilent Technologies, Palo Alto, CA, USA), provided with a quadrupole mass filter for mass spectrometric detection. GC separation was accomplished by a DB1-MS column (0.25 mm × 60 m, 0.5 μm film thickness, J&W; Agilent Technologies). The following chromatographic conditions were applied: injection of 1 μL volume, with a split ratio of 50:1 and injector temperature of 250 °C; the oven temperature program selected from 60 °C to 200 °C at 4 °C min^−1^ and then to 280 °C (5 min) at 50 °C min^−1^; and the He carrier gas constant flow set at 1.5 mL min^−1^ corresponding to a linear velocity of 32 cm s^−1^. The MS detector setting parameters were an electronic impact ionization mode set at 70 eV and the transfer line, source and quadrupole temperatures set at 300, 230, and 150 °C, respectively. For MS detection, the full scan mode was applied by selecting the mass range of 33–300 amu. The identification of EO constituents was performed by a comparison of the linear retention indices (LRI) and mass spectra of the chromatographic peaks with those obtained from a standard solution of pure reference compounds when commercially available (purchased from Merck, Sigma-Aldrich, Milan, Italy) (Appendix A). Linear retention indices were determined by analyzing, under the same conditions used for the EOs, a standard solution of C7–C30 saturated alkanes and according to the equation proposed by van den Dool and Kratz. When a pure compound was not available, tentative identification was based on the comparison of determined linear retention indices with those reported in the literature [55], in the NIST Chemistry WebBook database [56] and by a comparison of mass spectra with those reported in the NIST/EPA/NIH Mass Spectral Library (Version 2.4, 2020). The percent content of the compounds was determined from their peak areas in the GC Total Ion Current profile. Only compounds at a level higher than 0.05% were quantified. Each oil sample was analyzed in duplicate.

### 4.4. Enantioselective GC–MS Analysis of Volatiles in Dried Leaves and Flowers and EO Constituents

The enantioselective analysis of chiral compounds was performed on the EO constituents and also on volatiles isolated from the dried leaves and flowers to investigate the effects on enantiomeric distribution due to the distillation procedure. The isolation of volatiles from the dried leaves and flowers was carried out by the headspace solid-phase microextraction (HS-SPME) technique. Briefly, a weighted amount of died plant material (1.5 g) was placed in a 15 mL vial capped with a PTFE/silicone septum (Supelco; Sigma-Aldrich) for HS-SPME. The extraction was carried out by exposing a 2 cm, 50/30 μm DVB/CAR/PDMS fiber (Supelco; Sigma-Aldrich) to the headspace of the plant material for 30 min, while keeping the extraction temperature at 30 °C by a water bath. At the end of the extraction, the fiber was immediately inserted into the GC split–splitless injection port for the desorption step and the GC run was started. A duplicate extraction of each cheese sample was carried out. GC analyses were performed using the same GC–MS system reported above, equipped with a chiral column of Cyclosil-B (0.25 mm × 30 m, 0.25 μm of film thickness, stationary phase: 30% Heptakis (2,3-di-*O*-methyl-6-*O*-t-butyl dimethylsilyl)-β-cyclodextrin in DB-1701, J&W; Agilent Technologies). The chromatographic conditions were as follows: desorption for 5 min in the GC injector provided with a 0.75 mm glass liner suitable for SPME and operating at 240 °C in the spitless mode; oven temperature program from 55 °C (2 min) to 190 °C at 4 °C/min, then to 220 °C (5 min) at 20 °C/min; helium used as carrier gas at a constant flow of 1.5 mL min^−1^ corresponding to a linear velocity of 45 cm s^−1^; and MS detector parameters as stated above in Section 4.3. In the case of the analysis of the EO, 1 μL EO diluted in methanol (1:20 *v*/*v*) was injected and a slit ratio of 50:1 was set. All the other GC–MS conditions were the same as in the analysis of the HS-SPME isolates.

The identification of enantiomers was accomplished by a comparison with the mass spectra and linear retention indices (LRI) of authentic standards (Sigma-Aldrich). When authentic standards were not available, tentative identification was accomplished by a comparison with information reported in the NIST/EPA/NIH Mass Spectral Library (Version 2.4, 2020), in the literature [57] or kindly provided by Prof. Patrizia Rubiolo (personal communication). A standard solution of linear alkanes (C7–C30) was run under the same chromatographic conditions as the samples to determine the LRI of the detected compounds. Enantiomeric ratios were expressed as percent peak areas of the sum of the areas of the peaks of both enantiomers.

### 4.5. Indicator Microorganism Strains

The microorganisms used were *Salmonella enterica* serovar Typhimurium LT2 (DSMZ 18522; Braunschweig, Germany) and two isolates of *S. enterica* from chicken samples belonging to two different serovars (Derby and Give), provided by the Istituto Zooprofilattico Sperimentale del Mezzogiorno (Portici, Naples, Italy). The enterotoxigenic *Escherichia coli* strain K88 (ETEC, O149:K88ac) was provided by the Lombardia and Emilia Romagna Experimental Zootechnic Institute (Reggio Emilia, Italy). *Listeria monocytogenes* OH, *L. monocytogenes* CAL, *L. monocytogenes* SA and *L. innocua* 1770 were provided by the CREA-Research Centre for Animal Production and Aquaculture (Lodi, Italy), while *Pseudomonas putida* WSC358, *P. putida* KT2240 and *P. fluorescens* B13 were provided by Prof. Livia Leoni of Roma Tre University, Rome. With the exception of the ETEC growing in Luria-Bertani (LB) broth, Miller (DIFCO; Rodano (MI), Italy), all bacteria were routinely grown in tryptone soy broth (TSB; Oxoid, Basingstoke, UK) at their optimal growth temperature, which was 30 °C for the *Listeria* and *Pseudomonas* strains and 37 °C for the *Salmonella* and *Escherichia* strains.

### 4.6. Agar Spot Test

The spot-on-agar test was performed by spotting 2 μL EOs (CAR1, CAR2, CAR3, THY3, THY5) onto tryptone soy agar (TSA, 1.2%; Oxoid) plates previously seeded with 1 × 10^6^ CFU/mL indicator strains at the exponential growth phase. Spotted plates were then incubated for 18 h and inhibition zones (radii of the microbial growth inhibition, halos) around the wells were measured in millimeters (mm), accordingly with Balouiri et al. [58]. In the same plates, the test efficacy was confirmed by adding 2 μL of 50 μg/mL ampicillin (Sigma-Aldrich). Antimicrobial capacity was considered high (+++) when the radius value of the inhibition halo was greater than 7.5 mm, medium (++) when it was between 5 and 7.5 mm and low (+) when it was less than 5 mm.

### 4.7. Direct Contact Test

The direct contact test was performed as described by Serio et al. [32] and modified as follows. Stock emulsions containing 4% (*v*/*v*) EOs (CAR1 or THY5) were prepared in tryptone salt solution (8.5 g/L NaCl, 1.0 g/L tryptone; Oxoid) with 1% Tween 80 (Sigma-Aldrich) added as an emulsifying agent and vortexed for 10 min. The emulsions were filter-sterilized (0.22 μm filter pore size). To study the effect of different EO concentrations, appropriate aliquots of stock emulsions were diluted in 10 mL PBS, to obtain final concentrations of 0–0.12–0.25–0.50% EOs.

EO dilutions were inoculated with a standard bacterial suspension (1 × 10^8^ CFU/mL) of *L. monocytogenes* OH or *S.* Typhimurium LT2 indicator strains and incubated at 37 °C for 30 min. Bacterial cells were also exposed to 0% EO (control, C) and to 0% EO plus 0.125% Tween 80 with or without 50 μg/mL ampicillin, to confirm test efficacy and to exclude any antimicrobial effect of Tween 80, respectively (CTw + Amp or CTw). Bacteria were harvested by centrifugation for 10 min at 1100× *g* at 4 °C and appropriate dilutions were plated on tryptone soy agar to evaluate bacterial cell viability by colony counting, after the incubation of the plates for 18 h.

### 4.8. Challenge Test

Minced cow meat purchased at the supermarket was used to perform the challenge test. Meat portions (100 g each) were contaminated with *S.* Typhimurium LT2 or *L. monocytogenes* OH inocula (approximately 1 × 10^4^ CFU/g), according to Zinno et al. [59]. To ensure the proper distribution of the pathogens, the inoculated samples were homogenized in Stomacher bags (Bag Mixer-400; Interscience, France) for 2 min at room temperature. After homogenization, the inoculated meat portions were split in two equal quantities (50 g each), one of which was treated with 0.5% CAR1 or THY5. In addition, one meat portion was not inoculated, nor oil-treated (control). The Stomacher bags were wrapped and stored under aerobic conditions at 4 °C for up to seven days.

Microbiological analysis of the populations of *L. monocytogenes* OH and *S.* Typhimurium LT2 was carried out after 0–1–2–3 and 7 days of refrigerated storage. At each sampling time, the aliquots of minced meat were aseptically 10-fold diluted in 0.9% NaCl and homogenized for 2 min at room temperature in Stomacher bags. The resulting slurries were serially diluted and plated in duplicate on selective OXFORD and xylose lysine deoxycholate (XLD) agar (Merck, Darmstadt, Germany) for *L. monocytogenes* OH and *S.* Typhimurium LT2, respectively. Populations of both bacteria were determined and quantified by colony counting, after incubation for 24 h.

Before inoculation with the two pathogens, the minced meat was also examined for the presence of any bacteria by an estimation of the total viable counts (TVCs) on plate count agar (PCA; Oxoid) after 72 h of incubation at 25 °C.

### 4.9. Intestinal Caco-2 Cell Culture

Caco-2 cells, obtained from INSERM (Paris, France), were subcultured at a low density [60] and used between passages 90–105. Cells were routinely maintained at 37 °C in a 95% 5% air/CO_2_ atmosphere at 90% relative humidity in DMEM containing 25 mM glucose, 3.7 g/L NaHCO_3_, 4 mM stable L-glutamine, 1% non-essential amino acids, 1 × 10^5^ U/L penicillin and 100 mg/L streptomycin (all from Corning, Milan, Italy), supplemented with 10% heat-inactivated fetal bovine serum (Euroclone, Milan, Italy). Cells were seeded on polyethylene terephthalate permeable Transwell filters (Falcon™ 10.5 mm diameter, 0.4 μm pore size; Corning) at a density of 1 × 10^6^ cells/filter and cultured for 17–21 days to allow complete differentiation. The medium was changed 3 times a week.

### 4.10. Cell Monolayer Permeability Assessments

Cell monolayer permeability was assayed by measuring the transepithelial electrical resistance (TEER), through a Millicell electrical resistance voltmeter (Merck Millipore, Darmstadt, Germany) on cells differentiated on Transwell filters, according to Sambuy et al. [39]. Only cell monolayers with TEER values higher than 1300 Ohm × cm^2^ were used, as this value is indicative of correct cell differentiation, as identified in preliminary experiments. For the experiments, TEER was recorded every hour for up to 8 h, and then at 24 h, at the end of treatments. Cell permeability was also assessed at 24 h by measuring the paracellular passage of the phenol red marker, as reported by Monastra et al. [61]. Briefly, 0.5 mL of 1 mM phenol red was placed in the apical (AP) compartment of cell monolayers while 1 mL PBS containing Ca^++^ and Mg^++^ was placed in the basolateral (BL) compartment. After 1 h incubation at 37 °C, the BL medium was collected, injected with 0.1 N NaOH and read at 560 nm (Tecan Infinite M200 Microplate Reader; Tecan Italia, Milan, Italy) to determine the phenol red concentration. Phenol red apparent permeability (Papp) was calculated from the following formula: Ct × V_BL_/∆t ×·C_0_ × A, where V_BL_ is the volume of the BL compartment (cm^3^), A is the filter area (cm^2^), ∆t is the time interval (s), Ct is the phenol red concentration in the BL compartment at the end of the time interval and C_0_ is the phenol red concentration in the AP compartment at the beginning. Phenol red Papp values below 1 × 10^−6^ cm s^−1^ were considered indicative of intact monolayers [27]. Thus, 1 × 10^−6^ cm s^−1^ was set as the threshold value, irrespectively of the statistical significance among samples.

### 4.11. Impact of Oregano EOs (CAR1 and THY5) on Intestinal Barrier Integrity

CAR1 and THY 5 were tested on Caco-2 cells to assay their potential effect on monolayer integrity through TEER and phenol red Papp measurements.

Several EO concentrations were added for up to 24 h to the AP compartment of Caco-2 cells differentiated on Transwell filters. To avoid possible interference with the fetal bovine serum proteins, cell monolayers were kept in a serum-free medium 16 h before the assay. The concentrations tested were 0.1–0.05–0.03–0.025–0.02–0.01% (obtained from a stock solution, diluted by 1:20 in ethanol from the initial preparation). As the control, the cells were also apically treated with 2% ethanol, corresponding to the concentration contained in the higher EO dilution tested.

### 4.12. Pathogen Adhesion Assay to Caco-2 Cells

For the adhesion assay, Caco-2 cells were seeded and differentiated in 24-well plates (Becton Dickinson, Milan, Italy) at 1 × 10^6^ cells/well. Cells were placed in an antibiotic- and serum-free cell culture medium 16 h before the assay. On the day of the assay, overnight bacterial cultures of the pathogen indicator strains *L. monocytogenes* OH and *S.* Typhimurium LT2 were diluted 1:10 in appropriate media and grown for approximately 2 h up to the exponential growth phase. After monitoring the OD 600, bacterial cells were harvested by centrifugation at 5000× *g* for 10 min, resuspended in antibiotic- and serum-free cell culture medium and added to the cell monolayers at a concentration of 1 × 10^8^ CFU/well (approximately 100:1 bacteria-to-cell ratio). CAR1 or THY5 were apically added together with the different indicator strains at a 0.02% concentration, identified as the highest nontoxic one from the Caco-2 permeability tests. Co-cultures of bacteria and Caco-2 cells were incubated at 37 °C for 1.5 h. Non-adhering bacteria were then removed by 5 washes with Hanks’ Balanced Salt Solution (Corning) and cell monolayers were lysed with 1% Triton-X-100, according to Guantario et al. [62]. Adhering, viable bacterial cells were quantified by plating appropriate serial dilutions of Caco-2 lysates on different media. For *S.* Typhimurium LT2, plating was performed by inclusion in violet red bile glucose agar (VRBGA) medium, while for *L. monocytogenes*, OH cell lysates were plated on Oxford medium. Plates were incubated for 18 h.

### 4.13. Inflammation Induction by TNF-α and EO Treatment for Gene Expression Analysis of NF-kB Pathway

To assess the potential anti-inflammatory ability of CAR1 and THY5 EOs assayed in the present study, Caco-2 cells differentiated on Transwell filters were pre-incubated for 1 h with CAR1 or THY5 at the 0.02% concentration in the AP compartment. After media withdrawal, cells were challenged for an additional hour with the pro-inflammatory cytokine TNF-α (Invitrogen, Rodano, Milan, Italy), added to both the AP and BL compartments at 15 ng/mL. Other samples were treated only with EOs for 1 h or only with TNF-α for 1 h.

At the end of treatments, RNA was extracted from the Caco-2 cells by using miRNeasy Plus Mini Kit (Qiagen, Hilden, Germany). Nucleic acid concentration was determined by a NanoDrop^®^ ND-2000 UV–VIS spectrophotometer (Thermo Fisher Scientific, Waltham, MA, USA) at an OD of 260 nm. All samples had an OD 260/OD 280 ratio higher than 1.8, corresponding to 90–100% of pure RNA. A total of 1 μg of RNA was reverse-transcribed into cDNA using ReadyScript™ cDNA Synthesis Mix (Sigma-Aldrich). The quantification of gene expression was determined by real-time PCR with a 7500 Fast Real-Time PCR System (Applied Biosystems, Waltham, MA, USA), using RT^2^ SYBR^®^ Green ROX qPCR Mastermix (Qiagen). Data were collected using 7500 software v2.0.5 and given as the threshold cycle (Ct). Ct values for each target and housekeeping gene were obtained and their difference was calculated (ΔCt). Primer efficiencies for all tested genes were similar. The comparative calculation, ΔΔCt, was used to find the difference in the expression levels between the control and treated samples. Data are expressed as the mean of log2 of fold change (FC) with respect to the control.

The following target genes were analyzed: inhibitor of nuclear factor kappa B (IkB)α, cellular inhibitor of apoptosis (cIAP)2, interleukin (IL)-1 α, IL-6, IL-8, and glyceraldehyde-3-phosphate dehydrogenase (GAPDH) was used as the housekeeping gene. Primer sequences are reported in Appendix A.

### 4.14. Statistical Analysis

One-way ANOVA followed by a post-hoc Tukey’s HSD test were used to evaluate the statistical significance, after performing Shapiro–Wilk’s and Levene’s tests, to verify normality and homogeneity of variance, respectively. Any *p* values < 0.05 were considered statistically significant. In the figures, mean values with different superscript letters significantly differ. Principal component analysis (PCA) was performed on mean values of the EO constituents levels, expressed as percentages, after normalization by arcsine transformation. The statistical analyses were executed with Microsoft Office Excel 2011 upgraded with XLSTAT (ver. 4 March 2014).

## 5. Conclusions

The characterization of the chemical and biological properties of native plant species with highly functional, nutraceutical and health protection properties may be important to promote innovative products with many possible applications (food functional, phytocosmetics, phytotherapy, plant defense), while contributing to the preservation of agricultural biodiversity in the Mediterranean area. The two *Origanum vulgare* genotypes investigated in the present study were selected as they are widely cultivated in Sicily, but little information was available up to now on the biological properties of their EOs.

The investigations carried out in the present study demonstrated a strong antimicrobial effect of EOs from the two genotypes on Gram-positive and -negative bacteria, both in vitro and in a challenge test on a food matrix, suggesting their potential use as control agents against a wide spectrum of foodborne pathogens, with the view to looking for efficient alternative strategies for foodborne disease containment. The experiments performed on Caco-2 cells, a suitable and validated human intestinal in vitro model, indicated that the EOs displayed essentially comparable biological effects, as they were both able to reduce pathogen adhesion to Caco-2 cells at a 0.02% concentration without inducing an inflammatory state. To the best of our knowledge, this is the first report focusing on the effect of oregano EOs on intestinal permeability, an aspect poorly explored in the literature, that lays the groundwork for further investigations aimed at a more comprehensive understanding of safety of use in humans. As a whole, the marked diversity in the chemical profile of EOs from the two genotypes did not produce relevant differences in the considered biological activities. This suggests the need for additional studies to clarify the contribution of individual oregano EO constituents to the investigated biological activities, along with the elucidation of the role of minor compounds and their possible synergistic interactions with the major active constituents, to develop a model for predicting biological activity based on chemical composition. Finally, from a chemical perspective, the analysis of enantiomeric distribution was also confirmed to be a valuable tool for the recognition of geographical origin in the case of oregano EOs.

## Figures and Tables

**Figure 1 plants-12-01344-f001:**
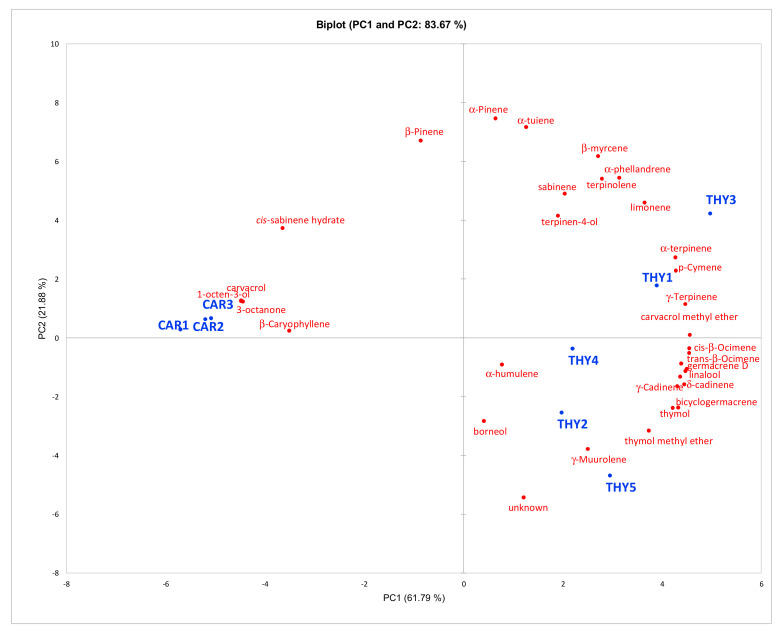
Principal component analysis (PCA) biplot of the first two PCs obtained in the dataset of the chemical compositions of EOs of the hybrid *Origanum vulgare ssp. viridulum* × *Origanum vulgare ssp. hirtum* (CAR1, CAR2, CAR3) and of the subspecies *Origanum heracleoticum* L. (THY1, THY2, THY3, THY4, THY5).

**Figure 2 plants-12-01344-f002:**
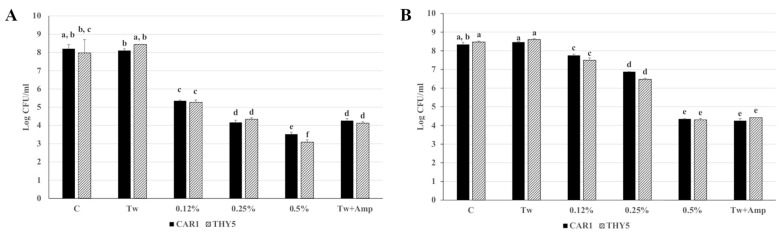
Direct contact test of CAR1 and THY5 antimicrobial activity (**A**) against *L. monocytogenes* OH and (**B**) *S.* Typhimurium LT2. Each EO was tested at 0.12, 0.25 and 0.5% concentrations. C: 0% EO; CTw: 0% EO + 0.125% Tween 80; CTw + Amp: 50 μg/mL ampicillin + 0.125% Tween 80. Bacterial cell viability is expressed as the geometric mean of CFU/mL ± SD of one experiment carried out in triplicate. Means without a common letter significantly differ, *p* < 0.05.

**Figure 3 plants-12-01344-f003:**
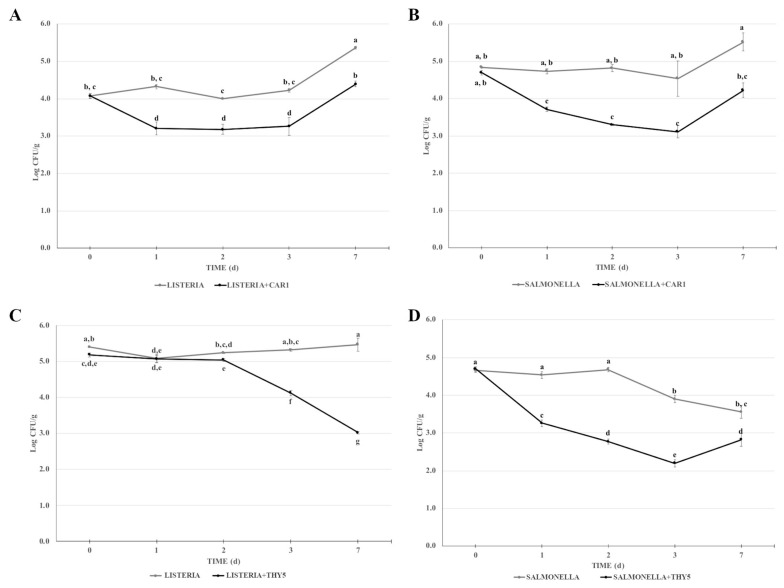
Challenge test of CAR1 against (**A**) *L. monocytogenes* OH and (**B**) *S*. Typhimurium LT2 and of THY5 against (**C**) *L. monocytogenes* OH and (**D**) *S*. Typhimurium LT2. The assay was performed in minced cow meat artificially contaminated with *L. monocytogenes* OH or *S*. Typhimurium LT2, with or without the addition of 0.5% CAR1 or THY5. The meat was stored at 4 °C for up to 7 days, homogenized at the indicated timepoints and appropriate dilutions were plated. Bacterial cell viability is expressed as the geometric mean of CFU/g ± SD of one experiment carried out in triplicate. Means without a common letter significantly differ, *p* < 0.05.

**Figure 4 plants-12-01344-f004:**
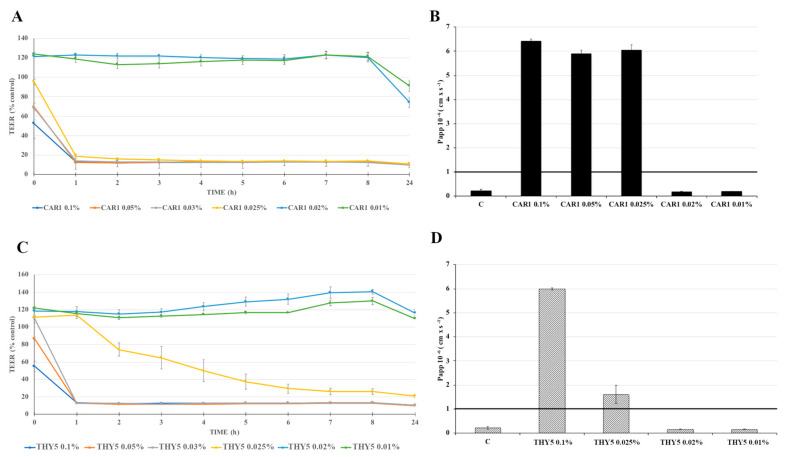
CAR1 and THY5 effects on Caco-2 cells monolayer integrity: Transepithelial electrical resistance (TEER) and phenol red apparent permeability (Papp). Cells were untreated (Control, C) or treated with different CAR1 or THY5 concentrations (0.01–0.1%). TEER values ((**A**) for CAR1 and (**C**) for THY5), recorded for up to 24 h, were calculated as the % of C for each timepoint. Phenol red Papp ((**B**) for CAR1 and (**D**) for THY5) was measured at 24 h and values are reported as cm s^−1^. A black line set at 1 × 10^−6^ cm s^−1^ represents the Papp threshold, indicating destroyed cell monolayer integrity for values above. Values represent the mean ± SD of the two independent experiments, carried out in triplicate.

**Figure 5 plants-12-01344-f005:**
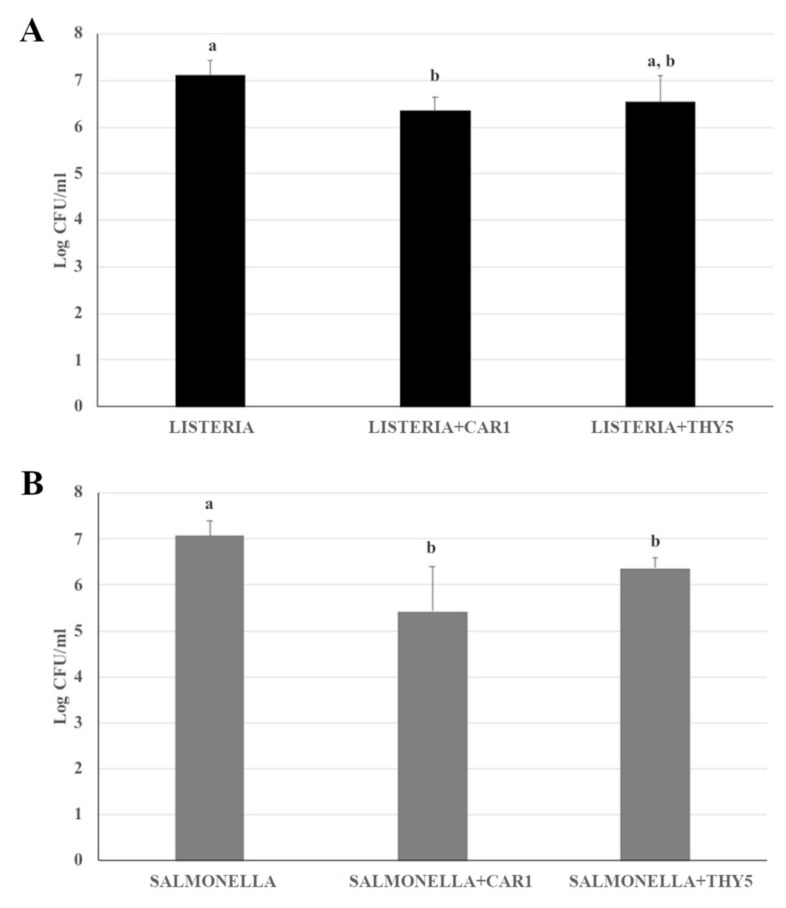
Adhesion reduction test for (**A**) *L. monocytogenes* OH and (**B**) *S*. Typhimurium LT2 in the presence of 0.02% CAR1 or THY5 in Caco-2 cells. Bacterial cell viability is expressed as the geometric mean of CFU/mL ± SD of the two independent experiments, carried out in triplicate. Means without a common letter significantly differ, *p* < 0.05.

**Figure 6 plants-12-01344-f006:**
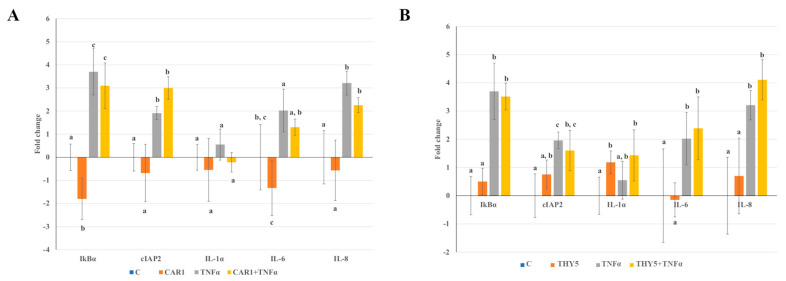
Study of NF-kB pathway gene expression in Caco-2 cells treated with (**A**) CAR1 and/or TNF-α and (**B**) THY5 and/or TNF-α. Caco-2 cells were pre-treated for 1 h with 0.02% CAR1 or THY5 and then treated for 1 h with 15 ng/mL TNF-α. IkBα, cIAP2, IL-1β, IL-6 and IL-8 gene expression was analyzed by real-time PCR. Data are expressed as the fold change (log2) ± SD of two independent experiments, carried out in triplicate. Means without a common letter significantly differ, *p* < 0.05.

**Table 1 plants-12-01344-t001:** Chemical compositions of the EOs of the hybrid *Origanum vulgare ssp. viridulum* × *Origanum vulgare ssp. hirtum* (CAR1, CAR2, CAR3) and of the subspecies *Origanum heracleoticum* L. (THY1, THY2, THY3, THY4, THY5). Data are expressed as the % content of individual constituents.

No.	Compound Name	Class	Oregano Subspecies/Hybrid							
			*Origanum vulgare ssp.* *viridulum* × *Origanum vulgare ssp.* *hirtum*			*Origanum heracleoticum* L.				
			CAR1	CAR2	CAR3	THY1	THY2	THY3	THY4	THY5
1	α-thujene	MH	0.74	0.75	0.78	1.04	0.72	1.19	0.83	0.35
2	α-pinene	MH	0.32	0.32	0.33	0.39	0.29	0.46	0.32	0.16
3	1-octen-3-ol	Ot	0.15	0.15	0.14	-	-	-	-	-
4	3-octanone	Ot	0.07	0.08	0.06	-	-	-	-	-
5	sabinene	MH	-	-	-	-	-	0.07	-	-
6	β-pinene	MH	0.05	0.06	0.06	0.07	-	0.08	-	-
7	β-myrcene	MH	1.15	1.18	1.19	1.63	1.22	1.84	1.41	1.01
8	α-phellandrene	MH	0.14	0.17	0.17	0.27	0.19	0.30	0.21	0.14
9	α-terpinene	MH	1.09	1.24	1.37	3.45	2.45	3.92	2.92	2.21
10	*p*-cymene	MH	2.64	2.73	2.59	4.79	4.11	5.25	4.63	3.60
11	limonene	MH	0.24	0.28	0.28	0.42	0.32	0.48	0.36	0.28
12	*cis*-β-ocimene	MH	-	-	-	1.44	1.30	1.87	1.36	1.64
13	*tr*-β-ocimene	MH	-	-	-	0.20	0.17	0.26	0.18	0.22
14	γ-terpinene	MH	5.33	6.30	7.11	17.96	13.16	22.13	15.90	16.95
15	*cis*-sabinene hydrate	OM	0.29	0.29	0.28	0.18	0.15	0.23	0.13	0.15
16	terpinolene	MH	-	-	-	0.07	-	0.08	-	-
17	linalool	OM	-	-	-	0.30	0.32	0.41	0.32	0.39
18	borneol	OM	-	0.08	-	-	0.44	-	-	-
19	terpinen-4-ol	OM	0.29	0.35	0.32	0.57	-	0.63	0.49	0.43
20	thymol methyl ether	OM	-	-	-	0.70	0.86	1.30	1.06	2.14
21	carvacrol methyl ether	OM	-	-	-	4.24	3.52	4.63	3.82	3.48
22	thymol	OM	0.14	3.19	1.68	56.22	65.47	47.32	61.35	59.36
23	carvacrol	OM	84.70	80.59	81.32	0.93	0.68	1.28	0.86	0.50
24	β-caryophyllene	SH	2.45	1.93	2.08	1.20	1.11	1.54	0.95	1.80
25	α-humulene	SH	0.06	0.08	0.09	0.08	0.08	0.11	-	0.13
26	γ-muurolene	SH	-	-	-	0.09	0.08	-	-	0.08
27	germacrene D	SH	-	-	-	1.52	1.36	2.25	1.23	2.22
28	bicyclogermacrene	SH	-	-	-	0.15	0.13	0.20	0.12	0.24
29	β-bisabolene	SH	0.15	0.24	0.16	1.28	1.17	1.52	1.02	1.56
30	γ-cadinene	SH	-	-	-	0.15	0.12	0.10	0.08	0.12
31	δ-cadinene	SH	-	-	-	0.47	0.39	0.39	0.33	0.47
32	unknown	SH	-	-	-	0.19	0.18	0.17	0.14	0.24
**Total content of compounds grouped in chemical classes**										
Monoterpene Hydrocarbons (MH)			11.71	13.03	13.88	31.74	23.94	37.93	28.11	26.58
Oxygenated Monoterpenes (OM)			85.42	84.50	83.59	63.13	71.44	55.80	68.02	66.46
Sesquiterpene Hydrocarbons (SH)			2.66	2.25	2.33	5.13	4.61	6.27	3.87	6.85
Others			0.22	0.23	0.20	-	-	-	-	-

**Table 2 plants-12-01344-t002:** (**a**) Enantiomeric distributions (%) of chiral compounds determined after isolation by HS-SPME from oregano dried leaves and flowers of the hybrid *Origanum vulgare ssp. viridulum* × *Origanum vulgare ssp. hirtum* (CAR1, CAR2, CAR3) and of the subspecies *Origanum heracleoticum* L. (THY1, THY2, THY3, THY4, THY5). (**b**) Enantiomeric distributions (%) of chiral compounds determined from EOs of the hybrid *Origanum vulgare ssp. viridulum* × *Origanum vulgare ssp. hirtum* (CAR1, CAR2, CAR3) and the subspecies *Origanum heracleoticum* L. (THY1, THY2, THY3, THY4, THY5).

Compound/ Enantiomer		LRI ^1^	Oregano Subspecies/Hybrid							
			*Origanum vulgare* *ssp. viridulum* × *Origanum vulgare ssp. hirtum*			*Origanum heracleoticum* L.				
			CAR1	CAR2	CAR3	THY1	THY2	THY3	THY4	THY5
(**a**)										
α-thujene	n.i. ^2^	950	65	64	64	65	65	65	65	64
	n.i.	952	35	36	36	35	35	35	35	36
α-pinene	S/−	984	12	12	12	9	9	9	9	10
	R/+	988	88	88	88	91	91	91	91	90
β-pinene	R/+	1026	73	73	72	73	74	74	74	73
	S/−	1032	27	27	28	27	26	26	26	27
α-phellandrene	R/−	1036	5	5	5	4	4	3	3	4
	S/+	1039	95	95	95	96	96	97	97	96
linalool	R/−	1217	89	90	83	90	92	91	92	90
	S/+	1224	11	10	17	10	8	9	8	10
terpinene-4-ol	S/+	1300	54	54	54	52	50	49	48	49
	R/−	1304	46	46	46	48	50	51	52	51
α-terpineol	R/+	1350	11	9	10	8	7	8	8	10
	S/−	1360	89	91	90	92	93	92	92	90
(**b**)										
α-thujene	n.i. ^2^	950	n.d.	n.d.	n.d.	n.d.	n.d.	n.d.	n.d.	n.d.
	n.i.	952	n.d.	n.d.	n.d.	n.d.	n.d.	n.d.	n.d.	n.d.
α-pinene	S/−	984	6	7	6	3	5	4	4	3
	R/+	988	94	93	94	97	95	96	96	97
β-pinene	R/+	1026	79	79	81	79	79	84	84	n.d.
	S/−	1032	21	21	19	21	21	16	16	n.d.
α-phellandrene	R/−	1036	n.d.	n.d.	n.d.	n.d.	n.d.	n.d.	n.d.	n.d.
	S/+	1039	n.d.	n.d.	n.d.	n.d.	n.d.	n.d.	n.d.	n.d.
linalool	R/−	1217	n.d.	n.d.	n.d.	n.d.	n.d.	n.d.	n.d.	n.d.
	S/+	1224	n.d.	n.d.	n.d.	n.d.	n.d.	n.d.	n.d.	n.d.
terpinene-4-ol	S/+	1300	58	59	58	63	62	61	63	62
	R/−	1304	42	41	42	37	38	39	37	38
α-terpineol	R/+	1350	0	0	0	0	0	0	0	n.d.
	S/−	1360	100	100	100	100	100	100	100	n.d.

^1^ Linear retention index determined from the Cyclosil-B column stationary phase (30% Heptakis (2,3-di-*O*-methyl-6-*O*-t-butyl dimethylsilyl)-β-cyclodextrin in DB-1701). ^2^ It was not possible to identify the two enantiomers due to the unavailability of standard pure compounds and lack of information from the literature.

**Table 3 plants-12-01344-t003:** Antimicrobial activity of oregano EOs against indicator pathogens (spot test).

Pathogen Indicator	Inhibition Halo Radius (mm)				
	CAR1	CAR2	CAR3	THY3	THY5
*Listeria monocytogenes* OH	+++	+++	+++	+++	+++
*Listeria monocytogenes* SA	+	+	+	+++	+++
*Listeria monocytogenes* CAL	+++	+++	+++	+++	+++
*Listeria innocua* 1770	++	++	++	+++	+++
*Salmonella enterica* Typhimurium LT2	+++	++	+++	+++	+++
*Salmonella enterica* Give	+++	+++	+++	+++	+++
*Salmonella enterica* Derby	+++	+++	+++	+++	+++
Enterotoxigenic *E. coli* (ETEC) K88	+++	+++	+++	+++	++
*Pseudomonas putida* WSC358	+++	++	+++	+++	++
*Pseudomonas putida* KT2240	+++	++	++	++	++
*Pseudomonas fluorescens* B13	+	+	+	++	++

Antimicrobial activity was considered high (+++) when the value of the inhibition halo radius (hr) was greater than 7.5 mm (hr > 7.5 mm), medium (++) when between 5 and 7.5 mm (5 mm ≤ hr ≤ 7.5 mm) and low (+) when less than 5 mm (hr < 5 mm).

## Data Availability

The data presented in this study are available upon request from the corresponding authors.

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
