# Peer review of "Chemical Composition and Biological Activities of Essential Oils from Origanum vulgare Genotypes Belonging to the Carvacrol and Thymol Chemotypes"

_plants, 2023, doi:10.3390/plants12061344_

Round 1

Reviewer 1 Report

The manuscript of Paola Zinno et al. “ Chemical composition and biological activities of essential oils from Origanum vulgare genotypes belonging to the carvacrol and thymol chemotypes” is dedicated to the analysis of essential oils and HS-SPME extracts from dried leaves and flowers of the hybrid two subspecies of Origanum vulgare and Origanum heracleoticum. The content of selected monoterpene hydrocarbons, oxygenated monoterpenes and sesquiterpene hydrocarbons and enentiomeric distribution of chiral compounds were tested. For selected EOs, antimicrobial activity against L. monocytogenes OH and S. Typhimurium LT2 was assessed both in vitro and in complex food matrix. The effect of selected OEs on Caco-2 cell permeability was also examined.

The manuscript is very interesting and structured correctly and provides interesting data on the composition and properties of Origanum vulgare genotypes grown in Southern Sicily. However, I believe that the  presentation of identifying the compounds contained in OE is too poor. The literature retention index ranges of analyzed components overlap, therefore I am unable to verify the correctness of this analysis. I suggest at least in Table S1 to add the molecular weights of the identified compounds in order to obtain a more complete view of the data.

Moreover in Notes of Table S1 – should be PC insteed PS.

Author Response

Dear Reviewer,

thank you for your comments. Please find enclosed our replies. 

Best regards.

Reviewer 2 Report

Thanks for the opportunity to review this research. The manuscript entitled Chemical composition and biological activities of essential oils from Origanum vulgare genotypes belonging to the carvacrol and thymol chemotypes” have described the chemical and biological properties of EOs from two Origanum vulgare genotypes to promote alternative and innovative applications in food and pharmaceutical products. The subject of the manuscript is topical, but I recommend the publishing of the paper after the necessary corrections.

1. The abstract should be beginning with a sentence about the background of concept and the aims as well as novelty of study should be mentions. Please improve.

2. Introduction: Check and format the citations in the whole manuscript.

3. Material and methods: The used methods are accurate.

4. Results and discussion: The results are clearly presented. General remark to the discussion: In my opinion, the discussion provided by authors is difficult to follow and verify due missing critical details.

5. The reference sources are not formatted as required by the journal.

I have only one recommendation to authors: please improve the conclusion of the manuscript and check the text for technical errors.

Author Response

(The authors gave the same response as above.)
